# Pathogenic *KRAS* variants disrupt structure and dynamics: Insights from integrated computational analyses

Saqib Ishaq[1,2,3☯]*, Aizaz Ali[4☯], Obaid Habib[1], Shabir Ahmad Usmani[2], Irshad Ur Rehman[3], Abdul Aziz[2], Yasir Ali[4,5]*, Ajaz Ahmad[6], Amin Ullah[7]*

**1** Guangdong Provincial Key Laboratory of System Biology and Synthetics Biology for Urogenital Tumors, School of Basic Medicine, Shenzhen University Medical School, Shenzhen University (SZU), Shenzhen, Guangdong, China, **2** Department of Computer Sciences and Bioinformatics, Khushal Khan Khattak University Karak, Karak, Pakistan, **3** Center of Biotechnology and Microbiology, University of Peshawar, Peshawar, Pakistan, **4** Department of Biotechnology, Abdul Wali Khan University Mardan, Mardan, KPK, Pakistan, **5** School of Medicine, the Chinese University of Hong Kong, Shenzhen, China, **6** Department of Clinical Pharmacy, College of Pharmacy, King Saud University, Riyadh, Saudi Arabia, **7** Department of Allied Health Sciences, Molecular Biology Laboratory, Iqra National University (INU) Peshawar, Peshawar, Pakistan

☯ These authors contributed equally to this work.
* yasirali@awkum.edu.pk (YA); saqib.ishaq@kkkuk.edu.pk (SI); aminbiotech7@gmail.com (AU)

## Abstract

### Background

*KRAS* is among the most frequently mutated oncogenes in pancreatic, colorectal, and lung cancers, yet the structural and dynamic mechanisms by which specific coding variants alter its function remain poorly understood. This study employs an extensive in-silico protocol to identify the most detrimental non-synonymous single nucleotide polymorphisms (nsSNPs) within the *KRAS* gene.

### Objective

To identify and describe pathogenic non-synonymous single nucleotide polymorphisms (nsSNPs) in *KRAS* and elucidate their atomistic effects on structure, stability, and potential oncogenic initiation.

### Methods

A total of 173 nsSNPs were screened operating an integrated computational workflow combining pathogenicity prediction, evolutionary conservation assay, high-resolution structural modeling, molecular docking, atomistic molecular dynamics simulations, post-translational modification mapping, and protein–protein interaction assessment.

### Results

Four high-impact variants (L79P, A130P, G138E, and F141L) were determined as the most deleterious. Simulations revealed distinct perturbations in conformational

**Data availability statement:** All the data are available in the manuscript.

**Funding:** The author(s) received no specific funding for this work.

**Competing interests:** The authors have declared that no competing interests exist.

stability (RMSD), residue flexibility (RMSF), hydrogen bonding patterns, and binding energetics compared with the wild type, signifying mutation-induced destabilization and potential impairment of *KRAS* regulatory function. Notably, these variants are primarily associated with colorectal, pancreatic, and lung cancers, underscoring their clinical significance.

## Conclusion

This integrative examination provides mechanistic insightinto how specific *KRAS* variations may prompt oncogenic activation. The identified alterations represent high-priority targets for experimental confirmation, illuminating the power of computational techniquesin linking sequence variationand functional consequence in cancer biology.

## 1. Introduction

Cancer is a leading global health challenge, contributing significantly to mortality rates worldwide [1]. Among the most frequently diagnosed cancers are breast, prostate, and lung cancer, which remain a primary concern in cancer research and treatment [2]. *KRAS*, originally discovered in the Kirsten rat sarcoma virus, encodes the p21 protein, a critical element in viral transformation [3]. *KRAS* proteins, located in the cell membrane, are anchored by an isoprene group at the C-terminus [4]. As part of the MAPK signaling pathway, *KRAS* functions as a GTPase enzyme, regulating the conversion between GTP and GDP molecules, thus controlling crucial cellular processes such as growth, motility, and metabolism [5]. Several studies have found that *KRAS* effectors affect cell interactions with their extracellular environment, which may also influence cell growth, motility, and metabolism [6,7]. *KRAS* mutations are prevalent in approximately 25% of all human cancers. These mutations disrupt normal signaling, promoting uncontrolled cell division [8]. The most common cancers associated with these mutations include pancreatic, colorectal, and lung cancers, with the COSMIC database reporting significant associations with missense mutations in *KRAS* [9–11]. A homozygous deletion of *KRAS* leads to cell metastasis and cancer tumor transformation in pancreatic epithelial adenocarcinoma (PDAC) [12]. Many cancers are closely associated with the proto-oncogenes, including cardio-facial-cutaneous syndrome, ductal carcinoma of the pancreas, leukemias, mucinous adenoma, and Noonan syndrome [13–16]. Mutation sites of the *KRAS* gene have to be considered to be an effective way to develop new cancer treatment schemes [17].

Clinically, KRAS mutations are associated with poor prognosis, tumor aggressiveness, and resistance to targeted therapies such as EGFR inhibitors in colorectal and lung cancers. Unlike other oncogenic drivers (BRAF, EGFR, PIK3CA), KRAS remains difficult to target directly due to its high GTP/GDP affinity and limited druggable pockets. Therefore, analyzing nonsynonymous SNPs (nsSNPs) in KRAS is essential, as these variants can alter protein conformation and activity, driving

oncogenic signaling. Understanding the structural impact of deleterious nsSNPs may thus guide precision medicine approaches for KRAS-driven cancers.Approximately 90% of human genetic variation arises from single-nucleotide polymorphisms (SNPs), which are genetic markers found every 200–300 base pairs in the human genome [18]. Around 0.5 million SNPs are located in the exonic regions of the genome [19]. Variations in these SNPs, particularly nonsynonymous SNPs (nsSNPs), can alter amino acids in coding regions, impacting protein structure, stability, and function. High-risk nsSNPs, which can cause mutations or impair protein function, are of particular interest [20,21]. Studies show that nsSNPs account for nearly half of the variations associated with hereditary genetic diseases. In cancer research, nsSNPs in oncogenes have drawn significant attention [22,23]. Furthermore, several nsSNPs have been implicated in autoimmune diseases, infections, and inflammatory conditions [24,25]. Recently, various studies have identified the potential of nsSNPs within diverse genes, including *TAGAP, TOX3, CCR6, CCL21, CTLA4*, MYB oncoproteins, and HRAS proteins, to adversely impact protein structure and function. This indicates a potential association of these nsSNPs with diseases such as breast cancer, rheumatoid arthritis, and other autoimmune disorders [26–30].Genes related to immunity are highly polymorphic, with many nsSNPs yet to be identified. The bioactivity of SNPs influences drug reaction sensitivity in the signaling pathways of commonly used immune suppressants such as glucocorticoids, mycophenolate mofetil, azathioprine, tacrolimus, cyclophosphamide, and methotrexate. Previous in silico studies on KRAS mutations were often limited to small datasets or single predictive algorithms, offering only partial insights into their functional impact. In contrast, this study combines multiple bioinformatics tools, structural modeling, and stability analyses to comprehensively evaluate deleterious nsSNPs in KRAS. By linking structural alterations with potential therapeutic relevance, it provides a deeper molecular understanding of KRAS missense variants than previously reported. This study employs various in-silico tools to identify deleterious nsSNPs in the human *KRAS* protein, determining their impact on protein structure and function. Additionally, this study explores potential therapeutic interventions using molecular drug targeting. The findings of this study could enhance our understanding of disease pathogenesis and open new avenues for developing novel treatment strategies.

## 2. Materials and methods

### 2.1 Scientific recruitment of the SNPs

This study utilized advanced bioinformatics tools to investigate the impact of SNPs on the structure and function of the *KRAS* protein. We retrieved SNPs data from the Ensemble Genome Browser database(Release 109, 2023) (https://useast.ensembl.org/Homo_sapiens/Gene/Summary?db=core;g=ENSG00000133703;r=12:25205246-25250936). We obtained the protein sequence from UniProt (ID: P01116, Release 2023, 04) (https://www.uniprot.org/), focusing on nsSNPs, which can alter the amino acid sequence and disrupt normal protein function. This study was carried out in several steps, for which a schematic flowchart is provided to illustrate the overall procedure in Fig 1.

### 2.2 Identification of deleterious nsSNPs in the *KRAS* gene

We used several bioinformatics tools to validate deleterious nsSNPs associated with the *KRAS* gene. The nsSNPs were selected from the NCBI SNP database (Build 155) and analyzed using), PROVEANv1.1 (Protein Variation Effect Analyzer) (http://provean.jcvi.org/seq_submit.php) nsSNPs with scores ≤ −2.5 were considered deleterious [31], SNPs & GOv1.0 (https://snps.biofold.org/snps-and-go/snps-and-go.html) probability ≥0.5 was considered disease-associated [32]. PhD-SNP v1.1 (Predictor of human Deleterious SNP) (https://snps.biofold.org/phd-snp/phd-snp.html) probability ≥0.5 classified as deleterious [33], PolyPhen-2v.2.2.2 (Polymorphism Phenotyping-2) (http://genetics.bwh.harvard.edu/pph2/) scores ≥0.85 indicated probably damaging, 0.15–0.85 possibly damaging, < 0.15 benign [34]. A total of 173 nsSNPs were retrieved from dbSNP (Build 155) and filtered based on predicted pathogenicity for analysis.These tools helped us determine the potential impact of the identified nsSNPs on the *KRAS* protein's structure and function.

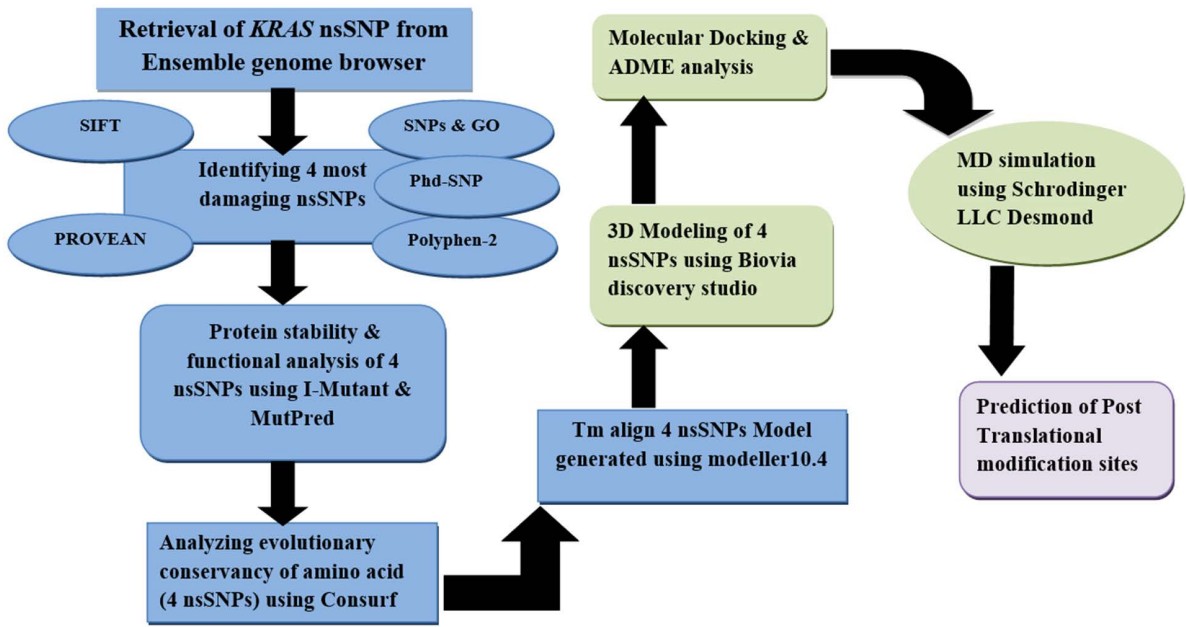

**Fig 1. A schematic flow showing the various steps involved in the bioinformatics analysis of the *KRAS* gene.**

## 2.3 Protein stability

We used two computational tools to assess the impact of nsSNPs on *KRAS* protein stability. I-Mutant 2.0 (http://folding.biofold.org/i-mutant/i-mutant2.0.html) [35] estimates the effect of SNPs on the free energy change (Delta Delta G: DDG) based on the protein's sequence or tertiary structure. DDG measures the change in Gibbs free energy, predicting the change in folding free energy by comparing the wild-type and mutant structures. Using I-Mutant 3.0, we classified mutations as substantially unstable ($\Delta\Delta G < -0.5$ kcal/mol), largely stable$\Delta\Delta G > 0.5$ kcal/mol), or neutral (($-0.5 \leq \Delta\Delta G \leq 0.5$ kcal/mol), predicting their impact on protein stability.

## 2.4 Functional analysis of *KRAS* gene

MutPred (http://mutpred.mutdb.org/) [36] is a computational tool that simulates alterations in the structure and function of wild-type and mutant sequences. It predicts the probability of gain or loss of these properties, presenting them as scores. The general score (g)≥0.5 indicates the likelihood of the amino acid substitution (AAS) associated with disease or harmful effects. Additionally, MutPred provides the top five property scores (p) as P-values, indicating the likelihood of affecting specific structural and functional properties.

## 2.5 Conservation analysis of *KRAS* nsSNPs

ConSurf (http://consurf.tau.ac.il/2016/) [37] is an advanced bioinformatics technique to evaluate the evolutionary conservation of amino acid locations within protein sequences. Conservation scores ranged from 1 (variable) to 9 (highly conserved). Positions scoring 8–9 were considered highly conserved and functionally important. This analysis provides insights into the degree of conservation of various nsSNPs, helping us understand their potential impact on protein function and structure.

## 2.6 3-D modeling of *KRAS*

The *KRAS* gene sequence identities range between 85% to 95% with its homologous structures, with an average identity of approximately 90%. In comparison, the query coverage range between 97% to 100% was high for these homologous

structures, indicating that almost the *KRAS* sequence was represented in the alignments. We used advanced computational tools to generate 3D models of wild-type and mutant *KRAS* proteins containing deleterious nsSNPs. Specifically, we used Modeller v10.4 [38] tools to create these models and Chimera v1.11 [39] to visualize and analyze the resulting structures. The TM-align (https://seq2fun.dcmb.med.umich.edu//TM-align/example/873772.html) algorithm compared the wild type and mutant protein structures, calculating the TM-score and root mean square deviation (RMSD) between the structures. A TM-score of 1 indicates a perfect match, while higher RMSD values indicate greater variation.

## 2.7 Energy minimization

We assessed the detrimental effects of nsSNPs on *KRAS* protein stability using the I-Mutant 2.0 tool (http://folding.biofold.org/i-mutant/i-mutant2.0.html). It predicts increases and decreases in protein stability. The altered protein sequences were analyzed with default parameters (Temp = 25°C, pH = 7). I-Mutant 2.0 results rely on the reliability index (RI), set from 0 to 10. Energy minimization of the 3-dimensional structures was performed using SWISS-PDB Viewer [40], comparing mutant and wild-type structures.

## 2.8 Molecular docking

Molecular docking was conducted employing AutoDock Vinav1.2.0 [41] to predict the binding affinities and interaction profiles of salirasib, sotorasib and garsorasib drug ligand with both wild and mutant *KRAS* proteins. Protein structure were retrieved for modeled utilizing Modeller v10.4 [42], followed by validation through Ramachandran plot assay to ensure structural accuracy. Prior to molecular docking and subsequent molecular dynamics simulations, GDP/GTP and $Mg^{2+}$ ions were removed from the *KRAS* structure to obtain the nucleotide-free (apo) form. Drug Ligands were accomplished from PubChem [43] and preprocessed by optimizing geometries using the MMFF94 force field in Open Babel. Charges were assigned applying the Gasteiger technique ensuring proper ligand. The Kollman charges on the protein were notated 0.936 for wild and −1.936 for mutant. The Gasteiger charges on the ligands for wild protein were −1.9969, 0.003 and −0.9964, while for mutant, the Gasteiger charges were −1.9965, 0.0035 and 0.9963. The grid box spacing was set to 1.000 Å for all the ligand bindings, with a size dimension of 26. The grid box center coordinates for the wild-type protein were x = 56.976, y = 57.222, z = 57.227, and for the mutant protein, the x = −57.013, y = −57.24, and z = −57.291, respectively. Grid box dimensions (26Å) and centers were chosen based on the *KRAS* GTP-binding site to targets residues pivotal for its oncogenicactivity. Docking exhaustiveness was set to 8 to balance computational efficiency and accuracy. Post-docking examination were executed using BIOVIA Discovery Studio [44] and PyMOL [45] to identify key interactions, including hydrogen bonds, hydrophobic contacts and π-π stacking, along with binding residue relevance to *KRAS* function.

## 2.9 ADME analysis of drug-likeness

The essential components of pharmacokinetics are absorption, distribution, metabolism, and excretion (ADME). Chemical structures and physiological parameters are correlated to identify pharmacokinetic features using chemical descriptors. An effective drug must be quickly absorbed, properly distributed, metabolized to maintain action, and safely eliminated. Following oral consumption, the blood-brain barrier (BBB) predicts blood transition into the brain. The skin permeability coefficient (log Kp) predicted the barrier's permeability. The drug's logP (octanol-water partition coefficient) indicated lipophilicity, affecting chemical metabolism and absorption. Hydrophobic drugs have a higher probability of binding to undesirable hydrophobic macromolecules, highlighting the importance of hydrophobicity. Initial ADME estimation during the discovery phase significantly lowers pharmacokinetics-related adverse events in clinical stages. The SwissADME server [46] uses experimental data and computational models to predict ADMET properties.

## 2.10 Molecular dynamics simulation

Molecular dynamic simulations (MDs) were performed for 100 nanoseconds using the Schrödinger LLC Desmond program. The initial step was docking the protein and ligand to determine the binding location within the target's active site

[47,48]. MD models simulate ligands' behavior over time, predicting binding status and replicating atom motions in a biological environment. We used Maestro's Protein Preparation Wizard to optimize, reduce, and compensate for missing residues in the ligand-receptor combination. The system was built using the System Builder tool. Using the OPLS_2005 force field and the TIP3P water model, the simulation was conducted at 310 K and 1 atm pressure [49–52]. The models were neutralized with 0.15 M sodium chloride and neutralizing ions to replicate physiological conditions. The models were equilibrated, and their development was recorded every 100 picoseconds.

### 2.11  PTM sites prediction

Protein function can be predicted through a thorough study of post-translational modifications (PTMs). Methylation sites in the *KRAS* protein were predicted using GPS-MSP 3.0 (http://msp.biocuckoo.org/) [53]. The prediction of phosphorylation sites at serine, tyrosine, and threonine residues was conducted using GPS 3.0 (http://gps.biocuckoo.cn/) and NetPhos 3.1 (https://services.healthtech.dtu.dk/services/NetPhos-3.1/). GPS 3.0 provided more specific results with a higher potential for phosphorylation than NetPhos 3.1. NetPhos 3.1(threshold ≥0.5) [54] utilized neural network ensembles with a threshold of 0.5. Residues with scores above this threshold were predicted to be phosphorylated. BDM-PUB (http://www.ubpred.org/) and UbPred (http://bdmpub.biocuckoo.org/prediction.php) were used to predict ubiquitylation sites in the *KRAS* protein. UbPred, with a balanced cut-off (cut-off ≥0.62), predicted lysine residues with scores equal to or higher than 0.62 as ubiquitinated.

## 3.  Results

### 3.1  Recruited nsSNPs

From the dbSNP database, the most comprehensive resource for SNPs, we recruited a total of 11,987 SNPs. Among these, 173 were nsSNPs, 94 were located in the 5' UTR, 1,264 in the 3' UTR, and the remaining were other SNPs as shown in Fig 2. Only the nsSNPs were selected for further analysis.

### 3.2  Identification of deleterious nsSNPs

All 173 nsSNPs recruited from dbSNP underwent analysis using four distinct bioinformatics tools to assess their impact on the function and structure of the *KRAS* protein. These in silico tools included PROVEAN, PhD SNP, SIFT, and SNP & GO.

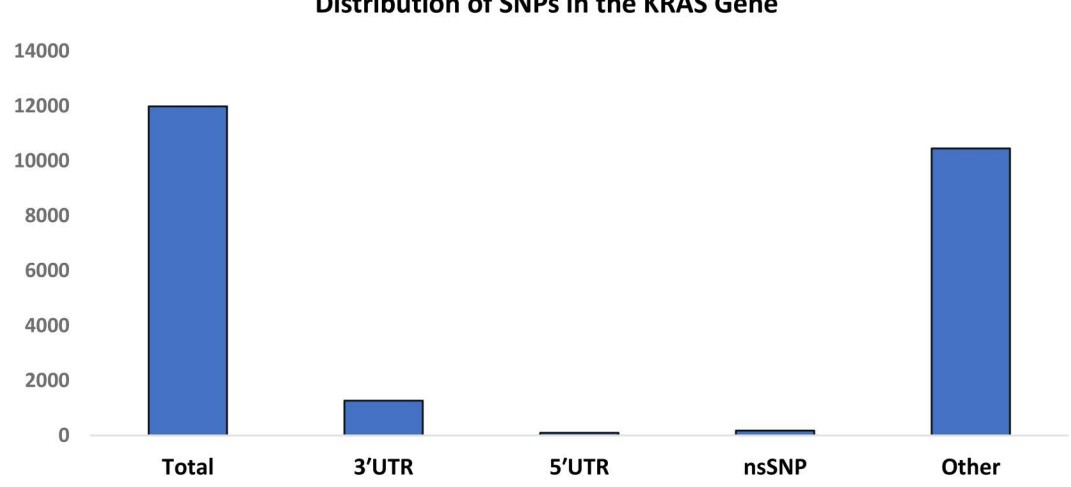

**Fig 2.  SNP distribution in the KRAS gene (absolute counts).** Bars represent the number of SNPs in each category.

In PROVEAN, a threshold value of −2.5 was set, below which variants were deemed harmful. According to PROVEAN results, 21 nsSNPs exhibited deleterious effects. SIFT utilized a Tolerance Index (TI) of 0.05 as its threshold value, identifying 15 nsSNPs as intolerant. Additionally, six nsSNPs were predicted to be disease-causing by PhD SNP, while SNP & GO labeled 11 nsSNPs as diseased. PolyPhen2 categorized results into benign, possibly damaging, and probably damaging, with the latter being the most confident prediction. Among *KRAS* nsSNPs, PolyPhen2 identified ten nsSNPs as probably damaging. We selected four nsSNPs deemed deleterious by all five tools for further analysis, as they are considered the most damaging illustrated in Table 1 and Fig 3.

### 3.3 MutPred prediction for structural and functional modification

The three nsSNPs selected by PolyPhen2 underwent analysis using the MutPred server. Results were provided with probability scores, as shown in Table 2. The predictions indicate that numerous nsSNPs have the potential to induce alterations in protein structure and may impact its function.

### 3.4 *KRAS* stability prediction and energy minimization

I-Mutant was employed to predict the stability of the *KRAS* protein concerning the selected nsSNPs and their respective amino acid substitutions. Each selected nsSNP was individually submitted, yielding stability results indicating either a decrease or increase, with a reliability index (RI) ranging from 0 to 10, as presented in Table 3. Among the four selected nsSNPs, none demonstrated an increase in stability; instead, all exhibited a decrease. This outcome suggests that these three nsSNPs may inflict more damage on the *KRAS* protein by reducing its stability.

**Table 1. Common most damaging nsSNPS in *KRAS* gene.**

| SNP ID | Amino acid | Provean | SNP&Go | | PHD-SNP | Polyphen-2 |
|---|---|---|---|---|---|---|
| | | Score | Probability | IR | Score | Score |
| **rs868857258** | L79P | −6.881 | 0.927 | 9 | 6 | 1 |
| **rs1463850736** | A130P | −3.248 | 0.852 | 7 | 6 | 1 |
| **rs754870563** | G138E | −5.915 | 0.759 | 5 | 1 | 1 |
| **rs138669124** | F141L | −4.534 | 0.737 | 5 | 1 | 0.091 |

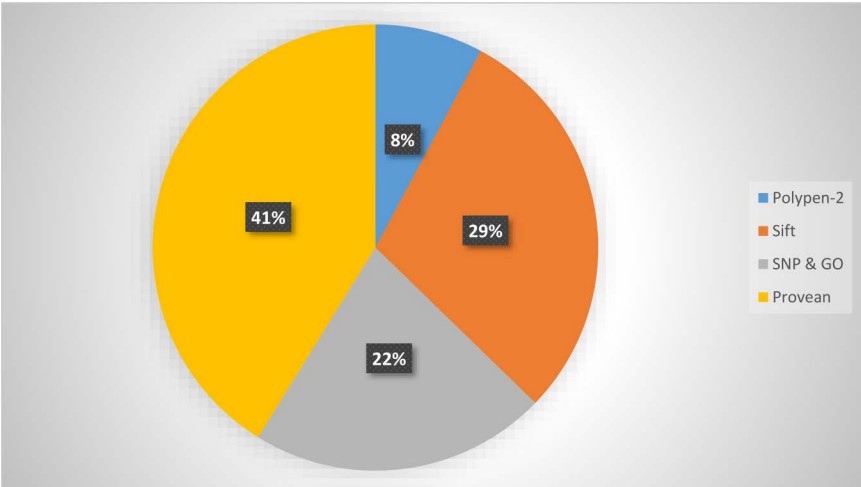

**Fig 3. Numbers (n) and percentage (%) of damaging nsSNP predicted by PROVEAN, SIFT, PhD SNP and SNP&GO.**

**Table 2. Probability values of deleterious nsSNPs identified in *KRAS*.**

| SNP ID | Amino Acid Change | Probability | Top Feature | Affected PROSITE and ELM Motifs |
|---|---|---|---|---|
| **rs868857258** | L79P | 0.967 | Gain of Allosteric site at F82 (P=1.6e-04) | ELME000328 |
| | | | Altered Ordered interface (P=3.4e-03) | |
| | | | Gain of Relative solvent accessibility (P=3.6e-03) | |
| | | | Gain of Catalytic site at E76 (P=1.7e-03) | |
| | | | Altered Metal binding (P=0.01) | |
| | | | Altered Stability (P=0.04) | |
| **rs1463850736** | A130P | 0.944 | Loss of Relative solvent accessibility (P=7.1e-03) | None |
| **rs754870563** | G138E | 0.868 | Altered Ordered interface (P=7.7e-03) | ELME000081, ELME000336 |
| | | | Altered Metal binding (P=0.02) | |
| | | | Altered DNA binding (P=0.02) | |
| | | | Gain of Catalytic site at E143 (P=0.04) | |
| | | | Altered Coiled coil (P=0.05) | |
| **rs138669124** | F141L | 0.940 | Loss of Relative solvent accessibility (P=6.1e-03) | None |

**Table 3. I-Mutant Result for the selected nsSNPs.**

| SNP ID | AMINO ACID | I-MUTANT STABILITY | RI |
|---|---|---|---|
| **rs868857258** | L79P | Decrease | 5 |
| **rs1463850736** | A130P | Decrease | 7 |
| **rs754870563** | G138E | Decrease | 6 |
| **rs138669124** | F141L | Decrease | 5 |

## 3.5 Energy minimization

The energy minimization results for both wild-type and mutant proteins indicate notable instability, reflecting significant differences between the two receptor types. These minimized energy values are crucial for understanding various aspects such as stability, reactivity, binding interactions, protein folding, and other functional properties. When comparing the minimized energy values of selected nsSNPs in both wild-type and mutant protein structures, the average mean values for bonds, angles, torsion, non-bonded interactions, electrostatic constraints, and the total residue value are shown in Table 4.

## 3.6 Evolutionary conservation of *KRAS* protein

ConSurf provided results for all amino acid residues in *KRAS*, but our primary focus was on the locations of the three identified nsSNPs. According to ConSurf predictions, G138E was predicted to be exposed and functionally significant. However, its low conservation score (2) suggests that mutations at this site are less likely to severely affect KRAS function compared to nsSNPs at highly conserved residues, which are predicted to have a greater impact on protein activity and stability. The structural residues L79P, A130P, and F141L were predicted to be buried. The conservation scores for each of the selected nsSNPs are detailed in Table 5. These revelations, as presented in S1 Fig, identify the importance of these residues in maintaining the biological function and structural integrity of *KRAS*. These findings indicate that nsSNPs located in highly conserved regions are the most detrimental to the function and structure of the *KRAS* protein.

## 3.7 3D-modelling of *KRAS* and its mutants

I-Mutant predicted a decrease in stability for four nsSNPs in the *KRAS* protein, making them prime candidates for final protein modeling. We submitted the protein sequences, including single amino acid variations for both the wild and mutants, to

**Table 4. Minimized energy values of selected nsSNPs in both wild and mutant protein structure.**

| SNP ID | Amino Acid | Bonds (Å) | Angles (°) | Torsion (kcal/mol) | Non-Bonded (kcal/mol) | Electrostatic constraint (kcal/mol) | Total energy minimization (kcal/mol) |
|---|---|---|---|---|---|---|---|
| **rs868857258** | L79 | 0.321 | 2.792 | 10.839 | −55.68 | −13.27 | −52.723 |
| | L79P | 6.907 | 12.969 | 26.311 | −36.11 | −35.32 | −14.505 |
| **rs1463850736** | A130 | 0.425 | 2.323 | 1.008 | −33.01 | −7.01 | −35.278 |
| | A130P | 3.043 | 28.271 | 16.328 | −14.59 | −30.84 | −3.529 |
| **rs754870563** | G138 | 0.608 | 5.726 | 1.843 | −10.66 | −27.82 | −25.952 |
| | G138E | 1.030 | 5.841 | 2.700 | −10.84 | −8.64 | −9.771 |
| **rs138669124** | F141 | 2.234 | 1.189 | 7.753 | −56.27 | −2.24 | −42.342 |
| | F141L | 0.770 | 0.917 | 6.441 | −25.81 | −0.13 | −16.620 |

**Table 5. Conservation profiles of the selected nsSNPs.**

| SNP ID | Amino acid Change | Conservation Score | Prediction |
|---|---|---|---|
| **rs868857258** | L79P | 5 | Buried |
| **rs1463850736** | A130P | 6 | Buried |
| **rs754870563** | G138E | 2 | Exposed |
| **rs138669124** | F141L | 4 | Buried |

Mode lwild-type and mutant *KRAS* varied tool for protein structure prediction. Modeller 10.4 generated five models for each *KRAS* variant, both wild-type and mutant. These models were then analyzed using TM-align to calculate each mutant's TM-score and RMSD values, as shown in Table 6. The TM-score assesses structural similarity between the wild-type and mutant proteins, while the RMSD value measures the degree of structural deviation.All mutants showed TM-scores above 0.94, confirming preservation of the overall *KRAS* fold. RMSD values of 1.0–1.7 Å indicate only minor local deviations, reflecting limited conformational adjustments rather than major structural disruption.The structural validation of the modeled wild and mutant protein structures exploiting ERRAT and Ramachandran plot assessment demonstrated high quality and reliability. ERRAT scores of 88.9% and 88.8% were accomplished for the wild and mutant models, respectively, while Ramachandran plot examination unveiled 82.5% and 81.8% of residues in most favored regions confirming their suitability for subsequent analyses as depicted in S2 Fig. For visualization and molecular characterization, Chimera 1.11 was utilized to examine the protein structures, as depicted in Fig 4. This detailed analysis provides valuable insights into the structural impact of nsSNPs on the *KRAS* protein.Structural mapping showed that the identified variants are located within key functional regions of *KRAS*. L79P lies near the switch II region affecting GTP hydrolysis, A130P and G138E occur in the α4–β6 loop near the GTP-binding site, and F141L is close to the C-terminal region involved in membrane association. Their presence in conserved functional domains supports their destabilizing and oncogenic potential.

**Table 6. TM-score and RMSD values of 4 selected nsSNPs.**

| SNP ID | Residual Change | TM Score | RMSD Values |
|---|---|---|---|
| **rs868857258** | L79P | 0.96056 | 1.39 |
| **rs1463850736** | A130P | 0.96532 | 1.08 |
| **rs754870563** | G138E | 0.94942 | 1.73 |
| **rs138669124** | F141L | 0.96430 | 1.41 |

0.0 < TM-score < 0.30, random structural similarity 0 ± 0.3 and 0.5 < TM-score < 1.00, in about the same fold 0.5 ± 1.

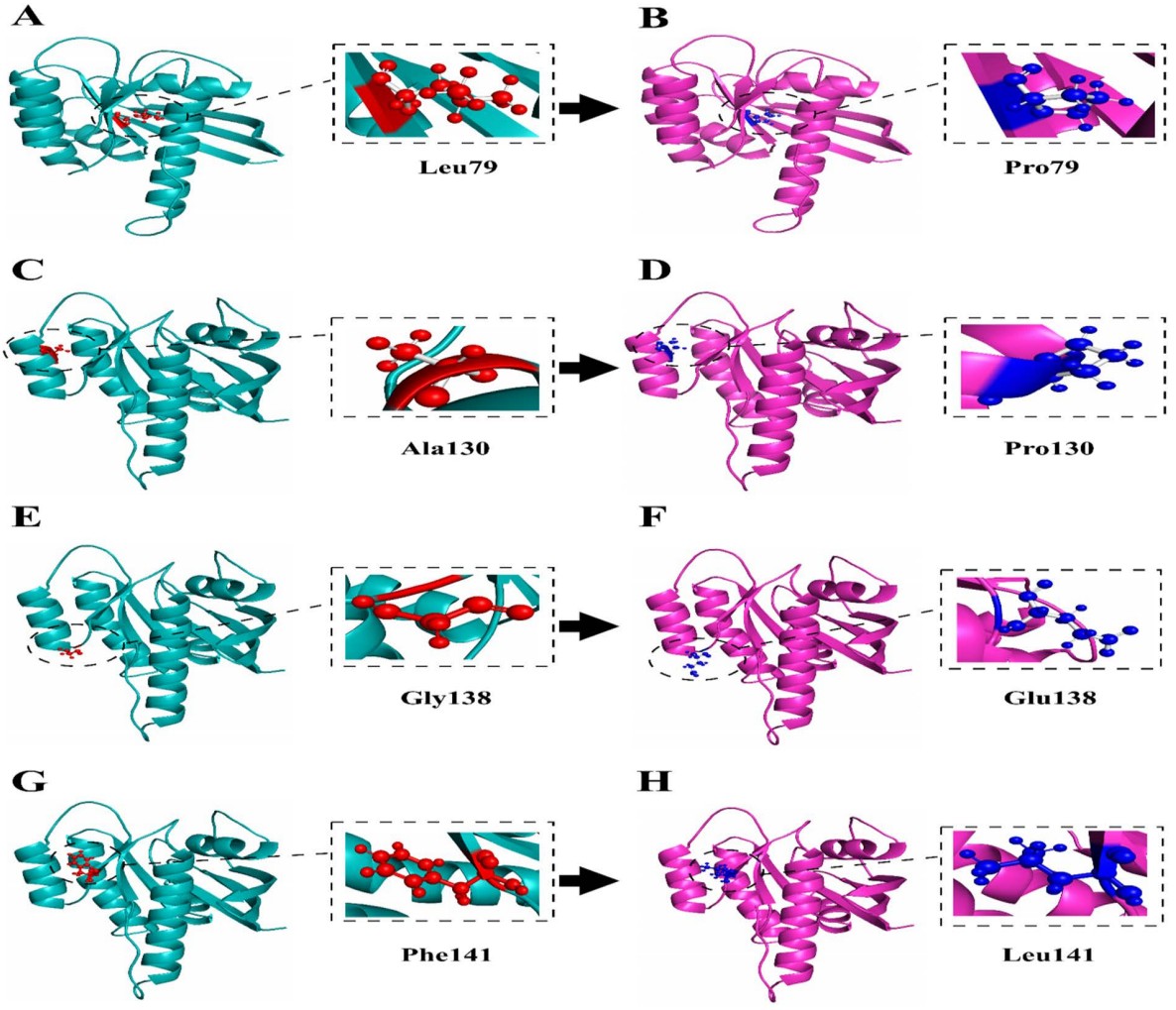

**Fig 4. (3D) Modeling of *KRAS* Wild type and Mutant Protein. (A-B)** L79P, **(C-D)** A130P, **(E-F)** G138E, **(G-H)** F141L.

### 3.8 Molecular docking

Molecular docking of the drug ligands salirasib (MW 358.5 g/mol), sotorasib (MW 560.6 g/mol) and garsorasib (MW 598.6 g/mol) with both wild and mutant *KRAS* proteins reveled expressive differences in binding affinities and interaction. The docking scores for the wild *KRAS* were salirasib (−6.0 kcal/mol), sotorasib (−5.4 kcal/mol) and garsorasib (−5.0 kcal/mol). For the mutant *KRAS*, salirasib exhibited the strongest binding affinity (−6.3 kcal/mol), followed by sotorasib (−5.2 kcal/mol) and garsorasib (−5.1 kcal/mol). This consequence intimate that salirasib is the most promising ligand for both wild and mutant *KRAS* proteins with its stronger binding to the mutant form potentially indicting greater therapeutic efficacy against *KRAS*-driven cancers. Key interactions between the drug ligands and *KRAS* residues were consistent. In the wild *KRAS* protein, Fig 5A salirasib interacted with Ala134, Arg135 and Met189, while Fig 5C sotorasibformedinteractions with Arg135, Pro140, and Arg151 and Fig 5E garsorasib with Ile142, Arg151 and Asp154. These resides are involved in main-taining the protein active conformation, inferring that these ligands may disrupt *KRAS* activity by binding to crucial sites. In the mutant *KRAS* protein, the docking poses indicated additional interactions, reflecting structural changes induced

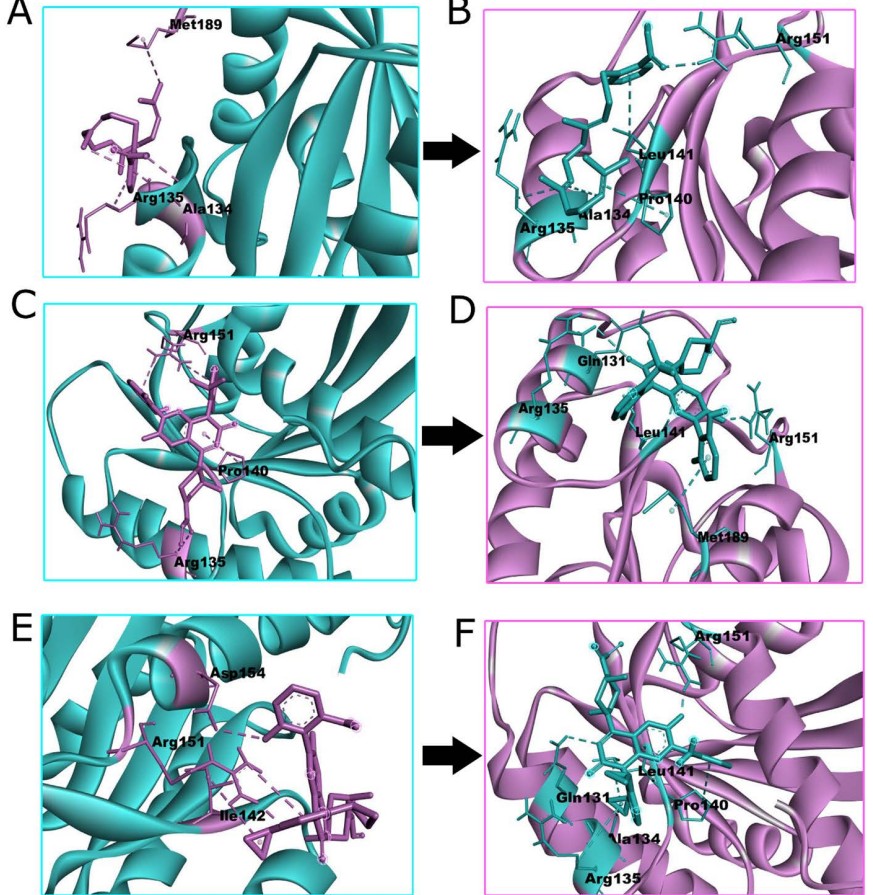

**Fig 5. Visualization of the Best Docked *KRAS* Gene Complex Wild and Mutant Protein (A, C, E) Cyan color represents wild (B, D, F) purple color represents mutant.**

by the mutation. Fig 5B Salirasib formed interactions with Ala134, Arg135, Pro140, Leu141 and Arg151. Fig 5D Sotorasib interacted with Gln131, Arg135, Leu141, Arg151 and Met189, while Fig 5F garsorasib engaged with Gln131, Ala134, Arg135, Pro140, Leu141 and Arg151. S1 Table reprised the docking scores, interaction residues and interaction distances for wild and mutant protein-ligand complexes, Illuminating key differences in binding affinities and interaction dynamics. The ligands salirasib, sotorasib, and garsorasib were geometry-optimized using the MMFF94 force field, with electrostatic analysis confirming proper charge assignment and reliable parameterization for docking and simulation studies.The 2D S3 Fig interaction illustrate key molecular interactions, including van der waals, alkyl hydrogen bonds and Pi-alkyl, offering a complementary perspective to the 3D structural exploration of protein-ligand biding. These mutation-specific variations in binding residues emphasize potential alterations in ligand recognition and could inform the design of targeted therapies for mutant *KRAS*. The docking poses were validated with an RMSD of 0.000Å, demonstrating the accuracy and reproducibility of the docking evaluation. These explorations enhance the potential of salirasib as a therapeutic agent, particularly for targeting the mutant *KRAS* form and refers that the selected ligands could be further explored in preclinical models for efficacy testing.These computational findings align with experimental studies showing salirasib's efficacy in inhibiting *KRAS*-driven signaling. The strong binding affinity and key residue interactions observed provide a mechanistic explanation for its activity, supporting further preclinical evaluation of salirasib and related inhibitors.

## 3.9 ADME analysis of a drug- likeness

The pharmacokinetic properties of the selected molecular compounds were evaluated by calculating various parameters, including aqueous solubility (log S), skin permeability coefficient (log Kp), drug-likeness assessment, molar refractivity (MR), permeability glycoprotein (P-gp) interaction, human oral absorption within the gastrointestinal tract (GI), and blood-brain barrier (BBB) permeation. Highly hydrophobic compounds tend to dissolve in fat globules, resulting in limited solubility in the stomach. As shown in Table 7, all the compounds exhibit drug-like properties due to their high solubility and absorption in the gastrointestinal tract (GI) except garsorasib has low GI absorpation. However, the investigation revealed that none of the compounds could penetrate the blood-brain barrier (BBB). Lipinski's Rule of 5, a widely used method for assessing drug-like properties, focuses on the solubility and permeability of compounds. Lipinski's Rule of 5 was employed to evaluate the hit compounds, analyzing their physical and chemical characteristics to determine their potential as orally active medications in humans. sotorasib and garsorasib showed at least one violation of molecular weight greater than 500 of Lipinski's rule. Despite this, these molecules can undergo chemical modification to enhance and improve their drug-like properties.

## 3.10 Molecular dynamic simulation

Following docking analysis, salirasib and sotorasib were prioritized for molecular dynamics simulations due to their superior binding affinities and consistent interaction stability with both wild-type and mutant KRAS proteins, indicating their potential as more robust inhibitors. Fig 6A and 6B depict the RMSD fluctuations of the carbon alpha atoms in the drug ligand-bound wild and mutant *KRAS* proteins, respectively. In the wild *KRAS* protein Fig 6A, stability was reached at ~10

**Table 7. ADME analysis using the Swiss ADME server.**

| S. N0 | C1 | C2 | C3 |
|---|---|---|---|
| Compounds (MolPortIDs) | CC(C)=CCC/C(C)=C/CC/C(C)=C/CSc1ccccc1C(=O)O | C=CC(=O)N1CCN(c2nc(=O)n(-c3c(C)ccnc3C(C)C)c3nc(-c4c(O)cccc4F)c(F)cc23)[C@@H](C)C1 | C=CC(=O)N1C[C@H](C)N(c2nc(=O)n(-c3c(C4CC4)ncnc3C3CC3)c3nc(-c4c(N)cccc4F)c(F)cc23)C[C@H]1C |
| H-BondAcceptors | 2 | 8 | 8 |
| H-BondDonors | 1 | 1 | 1 |
| TPSA(Å2) | 62.60 | 104.45 | 103.13 |
| ConsensusLogP$_{o/w}$ | 6.00 | 4.19 | 3.92 |
| MolarRefractivity | 111.00 | 160.26 | 170.63 |
| GI Absorption | High | High | Low |
| LogS | −3.70 Moderatly soluble | −3.82 Moderately soluble | −3.38 Moderately soluble |
| BBB Permeant | No | No | No |
| P-gpSubstrate | Yes | Yes | Yes |
| Log Kp (skinpermeation)cm/s | −3.40 cm/s | −6.90 cm/s | −7.79 cm/s |
| Drug likenessbased onLipinskirule | No violation | Yes;1violation (MW > 500) | Yes;1violation (MW > 500) |
| BioavailabilityScore | 0.85 | 0.55 | 0.55 |
| PAINS (alert) | 0alert | 0alert | 0alert |
| Brenk(alert) | 1 alert:isolated-alkene | 1alert | 2 alerts: anilin |
| Syntheticaccessibility | 3.58 | 4.65 | 5.14 |

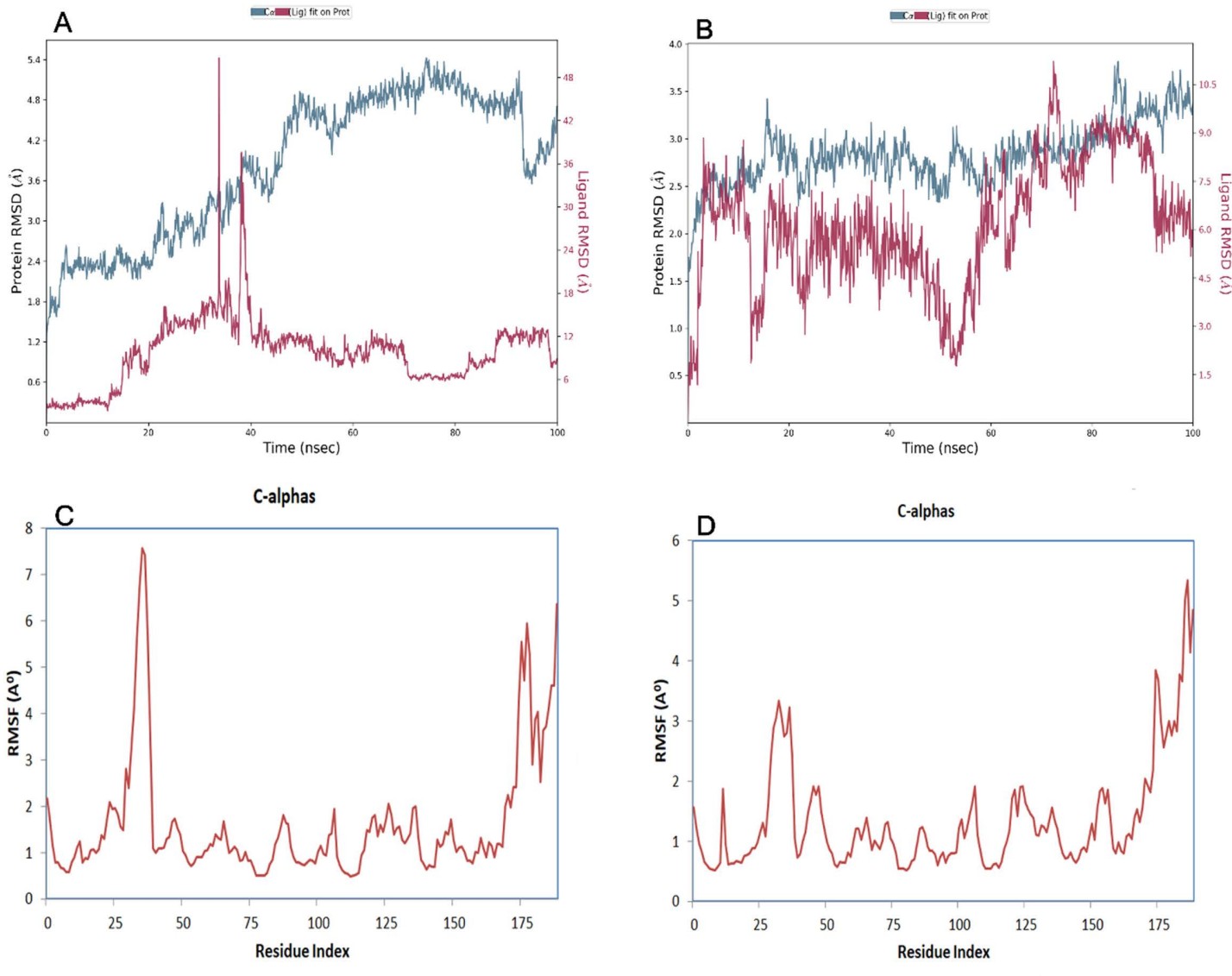

**Fig 6. Calculated RMSD values for alpha carbon (Ca) atoms (blue curves) of *KRAS* protein and protein fit with first ligands (red curves).** In the Fig 6, the (Fig 6A).Wild represents the blue, and red curves represent the ligand (6B). Mutants represent the blue, and ligands represent the red curve. Line representation of the evolution of root mean square fluctuation (RMSF) of *KRAS* Ca during the MD simulation (6C) Wild, (6D) Mutant.

ns and the RMSD fluctuated within a range of ~3.0Å throughout the simulation. Despite a slight increase in RMSD around 60 ns, the structure exhibited minimal fluctuation, indicating stability. The ligand remained stable after 40 ns with parodic fluctuations intimating limited shifts in biding, but it ultimately reached equilibrium by the end of the simulation. For the mutant *KRAS* protein, Fig 6B stability was achieved by 20 ns with RMSD values remaining consistent within ~1.0Å for the majority of the simulation. A tiny RMSD enhance was discerned at 80 ns, followed by minimal fluctuation between 80–90 ns signaling slight system acclimation before the structure stabilized again. The ligand in the mutant complex reached equilibrium by 10 ns and maintained a steady RMSD thereafter. Fig 6C and 6D RMSF assay illuminated distinct flexibility design between the wild and mutant proteins. The wild exhibited higher fluctuations at residues 45 and 175 (RMSF ~7Å),

indicating enhanced loop dynamics critical for function. The mutant demonstrates reduced flexibility with peaks at residues 25 and 175 (RMSF ~5Å), intimating mutation-induced stabilization. These explorations manifest the variant impact on protein dynamics and potential functional alterations.

The secondary structural assay reveled delicate differences between the wild and mutant *KRAS* protein structures as depicted in Fig 7E and 7F. In the wild structure, alpha helices and beta strands constituted 26.19% and 19.23% of the total secondary structural element (SSE) fraction, respectively, while in the mutant structure these shifted slightly to 25.16% and 19.11%. Despite the overall secondary structural element, percentages were identical these minor variations indicate the alteration may subtly affect protein stability and folding potentially influencing its biological function. Fig 7G and 7H emphasize the key hydrogen bonding interactions observed during the simulation. In the wild *KRAS* drug ligand complex, the primary hydrogen bonding sites were PHE_141 and GLN_131. In the mutant *KRAS* drug ligand complex,

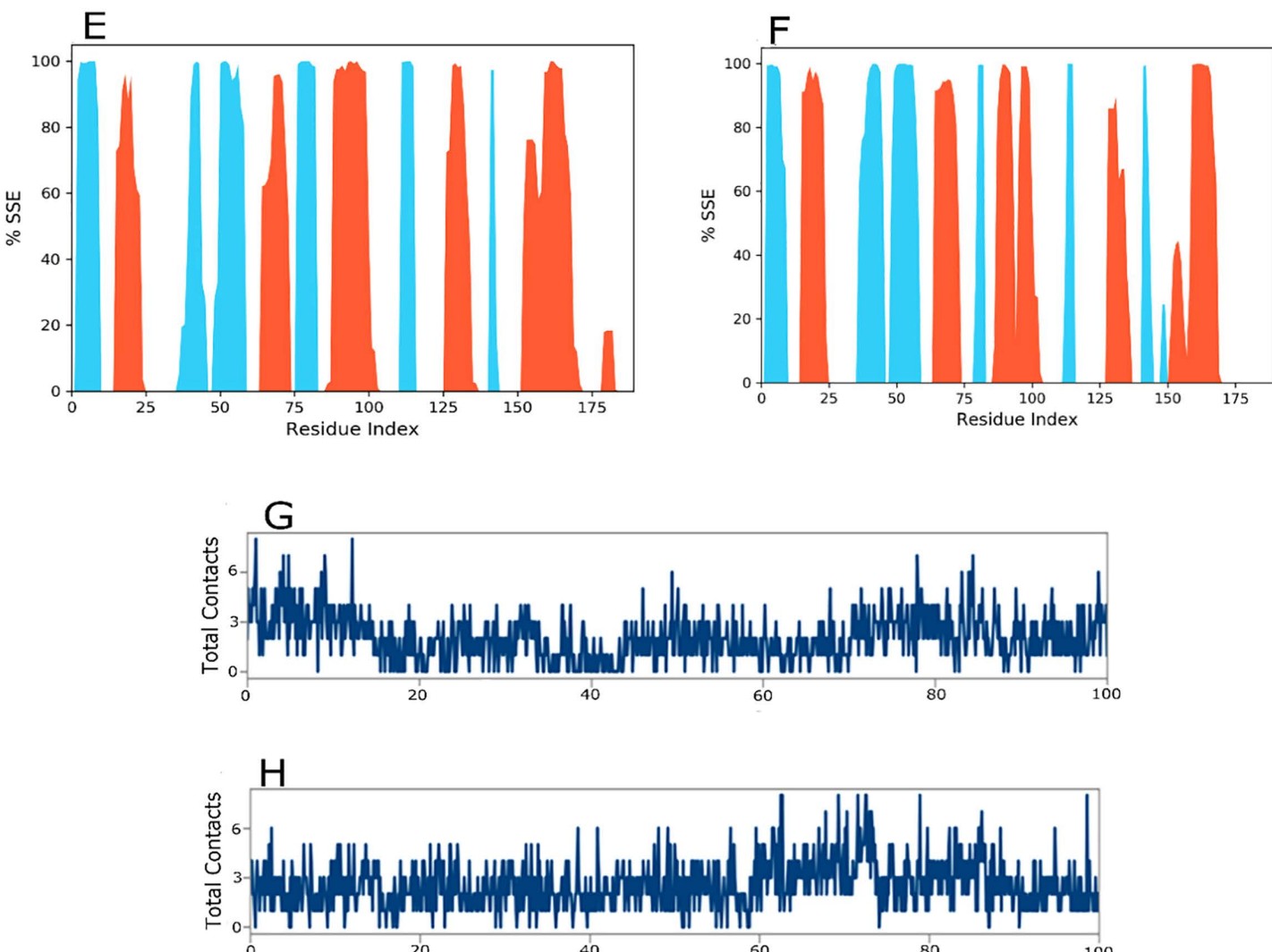

**Fig 7. Protein Secondary Structure element distribution by residue index throughout the protein structure with first ligand.** Red columns indicate alpha helices and blue columns indicate beta-strands. In the Fig 7, the (Fig 7E) Wild, (7F) Mutant. Timeline illustration of the protein-ligand interactions and contacts (H-bonds, ionic, hydrophobic, water bridges) with first ligand, (7G) Wild, (7H) Mutant.

the hydrogen boding residues shifted to LEU_141 and ILE_188. These variations in the hydrogen bonding network implies altered ligand interactions in the mutant form, potentially affecting the protein binding affinity and stability. Throughout the simulation, the protein drug ligand complex exhibited stable interactions as evidenced by the consistent number of contacts, including water bridges, ionic bonds, hydrophobic interactions and hydrogen bonds demonstrated in the interaction histograms.

The fluctuation of RMSD values for the alpha-carbon atoms of protiens bound to drug ligand over time is shown in Fig 8. In Fig 8A, the wild *KRAS* protein complex with the second ligand reached stability around 10 ns. After this stabilization, the RMSD fluctuated within a narrow range of 1.5Å, indicaing the overall stability of the protein drug ligand complex. The ligand reached equilibrium around 20 ns and remained stable for the remainder of the simulation. Moreover, intermittent fluctuaions in RMSD suggested occasional shifts in the binding state, although these differences did not significantly affect the drug ligand overall stability. The RMSD remained stable for approximately 100 ns, confirming the equilibrium of

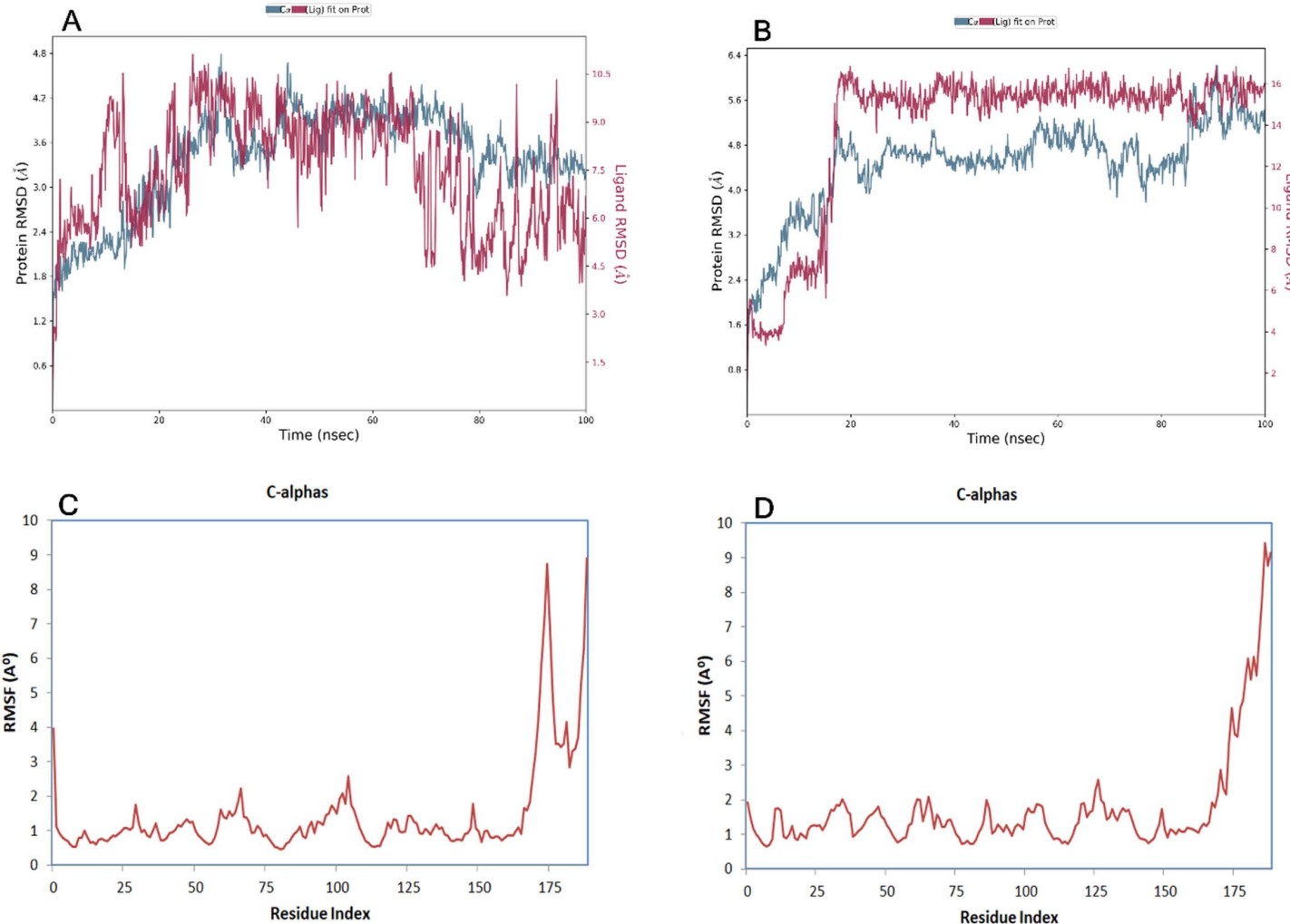

**Fig 8. Calculated RMSD values for alpha carbon (Ca) atoms (blue curves) of *KRAS* protein and protein fit with second ligands (red curves).** In the Fig 8, the (Fig 8A). Wild represents the (blue curves), and (red curves) represent the ligand. (8B) Mutant represents the blue, and ligand represents the red curve. Line representation of the evolution of RMSF of *KRAS* Ca during the MD simulation.(8C) Wild, (8D) Mutant.

the complex. In the instance of the variant strcture Fig 8B, RMSD stabilization was ascertained at 20 ns. After this point, RMSD fluctuations continue within a range of 1.0Å over the course of the 100 ns simulation. The drug acheived equilibrium at approximately 20 ns and maintained a consistent RMSD, indicating a stable binding interaction for the entire simulation period.

Fig 8C and 8D Root Mean Square Fluctuation (RMSF) examination reveals distinctive flexibility profiles for the wild and mutant proteins. The wild protein display higher fluctuations with peaks around residues 45 and 175 and a maximum RMSF of ~9Å, indicating dynamic loop regions. In contrast, the mutant protein unveils reduced flexibility with a maximum RMSF of ~5Å at residue 175, referring mutation-induced stabilization that may affect protein function.

The assay further identified loop regions and the N and C terminal as the primary sources of fluctuation in the profiles Fig 9E and 9F. These revelations point to variant induced difference in the flexibility of the protein structures. The reduced RMSF at the ligand binding sites supports the idea of stable and consistent protein drug ligand interactions. The distribution of secondary structural elements (SSEs) during the simulation is illustrated in Fig 9E and 9F. In the wild structures with the second drug complex Fig 9E, alpha helices and beta strands accounted for 22.53% and 19.41%, respectively resulting in an overall secondary structural elements (SSE) fraction of 41.94%. In the mutant structure with the second drug ligand Fig 9F, alpha helices and beta strands made up 25.02% and 17.01%, respectively subsequent in secondary structural elements (SSE) fraction of 42.03%. These minor variations in secondary structure percentages between the wild and mutant *KRAS* proteins may indicate variant induced changes in structural stability. Fig 9G and 9H highlight the hydrogen bonding interactions between the protein and drug ligand. In the wild complex, significant hydrogen boning occurred at residues ILE_139, PHE_141 and ARG_151, while in the variant complex, the key hydrogen bonding sites shifted to LEU_141, ARG_151 and ILE_142. These shits in hydrogen bonding residues intimate a variation dependent modification in drug ligand binding that could influence the protein function. Additionally, diverse types of interactions including as water bridges, hydrophobic contacts, ionic interactions and hydrogen bonds were adhered throughout the simulation. The cumulative count of these interactions over time is displayed in the histogram, reinforcing the sustained stability of the protein drug complex and identifying the relevance of these interactions in the biding process.

## 3.11 Predicted PTMs

PTMs regulate protein structures and functions, impact cell signaling and protein-protein interactions, and are pivotal in biological systems (Dai and Gu, 2010; Shiloh and Ziv, 2013). Our study explored whether selected nsSNPs induce alterations in the *KRAS* protein's PTMs. Various bioinformatics tools were employed to predict PTM sites in our target protein. Mechanistic interpretations are highlighted below to indicate potential functional consequences of the nsSNPs.

### 3.11.1 Methylation.
Methylation, an essential PTM, can influence DNA binding and gene expression in certain proteins when lysine residues undergo methylation. Using GPS-MSP 3.0, no methylation sites were predicted in *KRAS*. Although no methylation sites were identified, nsSNPs in KRAS could indirectly alter local structural features that affect methyltransferase accessibility in interacting proteins, potentially modulating downstream signaling pathways.

### 3.11.2 Phosphorylation.
Phosphorylation, another critical regulatory mechanism for proteins, serves as a molecular switch, modulating protein conformation, activation, deactivation, and signal transduction pathways (Deutscher and Sajer, 2005; Puttick et al., 2008; Ciesla et al., 2011; Sawicka and Sieser, 2014). Predictions from GPS 3.0 indicated 11 residues with phosphorylation potential, with a distribution of Serine (29%), Threonine (42%), and Tyrosine (29%). Conversely, NetPhos 3.1 predicted 22 phosphorylation-capable residues. A comparison between GPS 3.0 and NetPhos 3.1 results identified common residues, detailed in Table 8. Variants L79P and F141L are located near predicted phosphorylation sites. These substitutions may alter the local structural environment, potentially interfering with kinase recognition and phosphorylation efficiency, which could modify KRAS activation and downstream signaling cascades.

### 3.11.3 Ubiquitylation.
It serves as a mechanism for protein degradation and assists in DNA damage repair (Gallo *et al.,* 2017). The ubiquitylation of the *KRAS* protein was predicted using BDM-PUB and UbPred. BDM-PUB identified ten

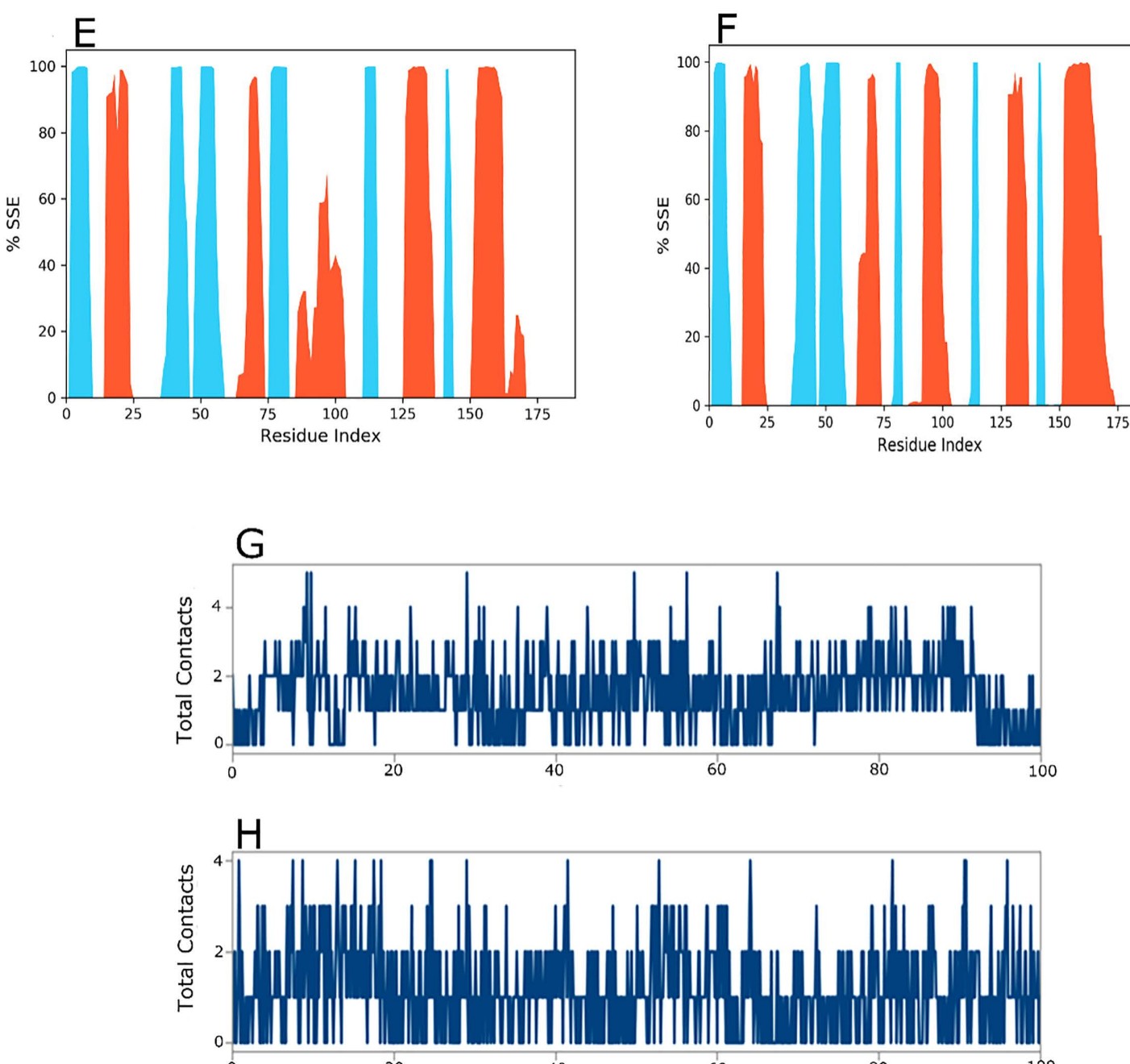

**Fig 9. Protein Secondary Structure element distribution by residue index throughout the protein structure with second ligand.** Red columns indicate alpha helices and blue columns indicate beta-strands. In the Fig 9, the (Fig 9E) wild and (9F) mutant. Timeline illustration of the protein-ligand interactions and contacts (H-bonds, ionic, hydrophobic, water bridges) with second ligand (9G) Wild, (9H) Mutant.

sites, while UbPred predicted one site in the lysine residue of the *KRAS* protein for ubiquitylation. The results of *KRAS* ubiquitylation predictions are presented in Table 9. The A130P variant lies adjacent to a predicted ubiquitination site. This substitution may hinder ubiquitin ligase recognition, potentially reducing KRAS degradation and increasing protein stability, which could contribute to sustained oncogenic signaling.

**Table 8. Prediction of phosphorylation sites in *KRAS* protein using GPS3.0 and NetPhos3.1.**

| Position | GPS3.0 | | | | NetPhos3.1 (Threshold 0.5) | |
|---|---|---|---|---|---|---|
| | Kinase | Score | Cutoff | | Kinase | Score |
| 39 | Atypical/PDHK/PDHK/PDK1 | 0.143 | 0.143 | | unsp | 0.971 |
| 65 | CAMK/PKD/PRKD2 | 2.244 | 1.986 | | Ckii | 0.502 |
| 89 | CAMK/CAMKL/MARK | 167.365 | 155.927 | | unsp | 0.996 |
| 106 | CAMK/PKD/PRKD2 | 2.254 | 1.986 | | Unsp | 0.604 |
| 172 | AGC/PKA | 24.871 | 21.268 | | unsp | 0.997 |
| 172 | AGC/PKC/PKCa/PRKCA | 19.73 | 17.673 | | Ckii | 0.518 |
| 2 | Atypical/Alpha/ChaK | 15.794 | 9.256 | | unsp | 0.543 |
| 50 | Other/NAK/BIKE/AAK1 | 4.821 | 2.856 | | Ckii | 0.502 |
| 144 | CK1/VRK/VRK2 | 40.98 | 39.717 | | Pkc | 0.554 |
| 158 | AGC/DMPK/GEK/DMPK | 3.182 | 2.819 | | unsp | 0.654 |
| 177 | AGC/GRK/BARK | 159.474 | 151.148 | | unsp | 0.884 |

**Table 9. Prediction of Ubiquitylation sites in *KRAS* protein using BDM-PUB and UbPred.**

| Position | BDM-PUB Score (Threshold 0.3) | UbPred Score (Threshold 0.62) |
|---|---|---|
| 5 | 1.01 | Not Ubiquitylated |
| 16 | 0.97 | Not Ubiquitylated |
| 101 | 0.69 | Not Ubiquitylated |
| 128 | 0.97 | Not Ubiquitylated |
| 169 | 1.59 | 0.36 |
| 173 | 1.01 | Not Ubiquitylated |
| 176 | 0.53 | Not Ubiquitylated |
| 182 | 0.50 | Not Ubiquitylated |
| 184 | 0.85 | Not Ubiquitylated |
| 185 | 1.36 | Not Ubiquitylated |

## 4. Discussion

This study aimed to utilize various computational tools to identify pathogenic nsSNPs within the *KRAS* gene. Previous research has extensively investigated the association between the *KRAS* gene and various diseases such as colon cancer, colorectal cancer, multiple types of tumors, and gastric cancer. KRAS has been linked to diseases such as colon and colorectal cancer. Computational studies show that receptor polymorphisms can affect protein function and disease susceptibility, offering insights into therapeutic targeting [55,56]. However, distinguishing functionally significant and damaging SNPs from neutral ones remains challenging. Given the potential importance of numerous SNPs within the *KRAS* gene, particularly in diseases like colon cancer, this study focused on analyzing SNPs in this gene region to assess their potential impact on *KRAS* protein function and regulatory mechanisms, which could be crucial in disease pathogenesis.

Five bioinformatics tools were employed to evaluate the effect of nsSNPs on the structure and function of the *KRAS* protein. Among these tools, PROVEAN predicted 41% of nsSNPs, SNP&GO 8%, SIFT 3%, and PolyPhen2 25.63% to be harmful or intolerant. Notably, four nsSNPs were consistently predicted as deleterious by all five tools.These nsSNPs (rs868857258, rs1463850736, rs754870563, rs138669124) are predicted to lead to premature stop codons, possibly truncating the *KRAS* protein. These findings are of particular significance because they represent specific mutations not

previously explored in the context of *KRAS*-related diseases.Moreover, comparing our identified variants (L79P, A130P, G138E, F141L) with well-characterized clinically reported KRAS mutations such as G12D, G12V, and Q61H underscores their potential clinical relevance. Like these classical oncogenic mutations, the novel variants may disrupt GTP hydrolysis, alter conformational stability, and affect effector interactions, suggesting possible contributions to KRAS-driven cancers and warranting further experimental validation.

In addition, we crosschecked these 4 nsSNPs in the Ensembl genome browser as other tools like CADD, REVEL, Mutation Assessor, and MetalR. According to CADD, REVEL, Mutation Assessor, and MetalR all four mutations (four nsSNPs) are deleterious and damaging. Additionally, we utilized an I-Mutant to assess the protein stability of these selected nsSNPs and their associated amino acid substitutions, revealing a decrease in protein stability for each nsSNP. These findings suggest that all four nsSNPs may be more damaging due to their destabilizing effect on the *KRAS* protein.Furthermore, ConSurf, which integrates evolutionary conservation data and solvent accessibility prediction, was employed to predict the conservation profile of the *KRAS* protein. Highly conserved residues are presumed to be functionally or structurally significant based on their location on the protein surface or core. In protein-protein interactions, crucial amino acids tend to be more conserved. ConSurf analysis provided insights into the potential effects of nsSNPs on the *KRAS* protein's conservation profile, offering a structural representation of *KRAS* protein alterations. Our verdicts propose that these mutations may contribute to disease progression by affecting the stability and function of *KRAS*, which has not been extensively reported in previous literature.The MutPred1.2 server is utilized to predict effects based on various features, including loss of methylation, gain of acetylation, altered ordered interface, changed disordered interface, and gain of intrinsic disorder. Among the mutations analyzed, L79P exhibits the highest P-value of 0.967, followed by A130P with P-values of 0.944 and 0.868, respectively. Conversely, mutation G138E demonstrates the lowest P-value of 0.868. These results imply potential structural and functional impacts of these nsSNPs on the *KRAS* protein. After protein modeling, our stability predictions were corroborated with those from the CUPSAT server (http://cupsat.tu-bs.de/), which assesses protein stability post-mutations based on structural data. Our I-Mutant predictions align entirely with CUPSAT forecasts, indicating heightened reliability of our predictive outcomes.The structural validation of the modeled wild and mutant protein structures exploiting ERRAT and Ramachandran plot assessment demonstrated high quality and reliability. ERRAT scores of 88.9% and 88.8% were accomplished for the wild and mutant models, respectively; while Ramachandran plot examination unveiled 82.5% and 81.8% of residues in most favored regions confirming their suitability for subsequent analyses.

Conventional homology or comparison modeling is widely employed in drug development due to its acknowledged accuracy [57]. This method assumes that each protein family component maintains a consistent fold, determined by a core structure resilient to sequence alterations [58]. SWISSPDB was employed to minimize three-dimensional structure energy to optimize normal and mutant *KRAS* protein structures. The process confirmed stability for both variants. Total energy values acquired for normal and mutant *KRAS* proteins were as follows: L79 (−52.723), L79P (−14.505), A130 (−35.278), A130P (−3.529), G138 (−25.952), G138E (−9.771), F141(−42.342), F141L (−16.620).The ADME dissection assessed fundamental pharmacokinetic properties of the selected compounds, including aqueous solubility, gastrointestinal absorption and drug-likeness. All compounds exhibited drug-like attribute with high solubility and GI absorption. However, none were able to penetrate the blood-brain barrier, indicting their suitability for targeting peripheral cancers. Lipinski's Rule of 5 reveled at least one violation for each compound, signifying potential areas for enhancement through chemical modifications. Despite this, these compounds remain promising candidates for further development with potential for enhanced drug-like properties through optimization. Lipinski's "Rule of Five" was employed to assess the "drug-likeness" [59], examining parameters such as the number of hydrogen bond donors (HBD), number of hydrogen bond acceptors (HBA), and calculated lipophilicity (log P) distributions. Evaluating these compounds' ADMET profiles is pivotal in determining their potential as anti-cancer medications, considering factors like absorption, distribution, metabolism, excretion, and toxicity (ADMET).Our molecular docking and MD simulations have provided valuable

insights for further medicinal investigation by elucidating potential ligand interactions [60,61]. Computational studies have shown that receptor polymorphisms can modulate small molecule binding and influence protein-ligand interactions, providing mechanistic insights for targeted drug development [62]. Additionally, we identified previously unexplored interaction dynamics between drug ligands and the wild and mutant *KRAS* proteins through molecular docking and MD simulation.

The complexities of binding affinity and interaction patterns between selected ligands and both the wild and nsSNPs within the *KRAS* gene structure were identified through molecular docking studies. RMSDis a critical quantitative technique to assess the stability of protein-ligand complexes by analyzing conformational changes in the protein backbone during MDS [63–65].

Similarly, RMSF is commonly utilized in MDS to gauge the flexibility of individual residues in a system [66,67]. Identifyinghydrogen bond interactions is fundamental for interpreting the complex molecular interaction patterns observed among protein-ligand complexes via MDS [68]. The structural comparison of docked molecules of normal and mutant type *KRAS* genes was conducted using BIOVIADiscovery Studio. These structures were docked with three top drug ligands: salirasib (M.W 358.5g/mol), sotorasib (M.W 560.6g/mol), and garsorasib (M.W 598.6g/mol). The best binding affinity docking scores for these ligands with both the wild-type and mutant *KRAS* proteins were noted, along with RMSD values (wild: −6.0 kcal/mol, −5.4 kcal/mol, −5.0 kcal/mol; mutant: −6.3 kcal/mol, −5.2 kcal/mol, −5.1 kcal/mol; RMSD 0.000). The docking results between receptor and ligand for all three drugs with wild-type and mutant structures are illustrated in Fig 5.The minor difference in docking energy between wild-type and mutant KRAS (−6.0 vs. −6.3 kcal/mol) suggests that salirasib maintains stable binding across both forms, indicating mutation-tolerant binding rather than a significant enhancement in affinity.Structural and dynamic characteristics of these significant biological complexes are clarified by MD simulations, revealing distinct stability profiles, flexibility patterns, and interaction dynamics between the wild-type and mutant structures of the *KRAS* gene.Although replicate simulations with different initial velocities were not performed, independent molecular dynamics simulations were conducted separately for each KRAS–ligand complex (wild-type and mutant with salirasib and sotorasib). The consistent convergence and stable RMSD/RMSF profiles across these independent systems support the reliability and reproducibility of the simulation outcomes. Future studies will incorporate multiple replicates to further strengthen statistical confidence.Though none of the predicted phosphorylation and ubiquitylation sites directly coincide with the most damaging nsSNPs, some variants are located nearby. These nsSNPs may affect kinase or ubiquitin ligase recognition, potentially altering KRAS activation, stability, and downstream signaling. Such changes could influence KRAS-driven oncogenic pathways and impact the efficacy of targeted therapies. Our study also highlighted four nsSNPs (rs868857258, rs1463850736, rs754870563, rs138669124) that may introduce premature stop codons, potentially resulting in truncated or inactive proteins.

## 5. Conclusion

This study presents a comprehensive in silico examination of (nsSNPs) in the *KRAS* gene, exploiting multiple computational server to distinguish potentiallypathogenic modifications. The innovation evasion in integrating protein stability forecasting, evolutionaryconservation assessment and drug-likeness interpretation to appraise their impact on *KRAS* protein function and therapeuticpotential. Four (nsSNPs) were consistentlypredicted as deleterious across all servers, emphasizing their potential aspect in diseasepathogenesis. Molecular docking and (MD) simulations in addition to identified favorable drug ligand interactions with both wild and mutant *KRAS*, enhanced opportunities for targeted therapeutic progression. While these computationalpredictionsprovide precious intuition, experimental validation detritus essential to confirm their clinicalrelevance. These findings extend precious implications for personalized medicine by discriminating specific *KRAS* variants that may consequence drug response and therapeutic design.Furthermore, the identified variants and ligand interactions provide a rational foundation for guiding structure-based drug design and prioritizing these mutations for future in vitro and in vivo functional assays.

## Supporting information

**S1 Fig. Consurf evaluation of KRAS nsSNPs illustrating critical regions for maintaining the protein functionality and structural stability.**
(DOCX)

**S2 Fig. Validation of wild and mutant KRAS protein models via ERRAT and Ramachandran plots.** (A-C) Wild, (B-D) Mutant.
(DOCX)

**S1 Table. Docking study for wild and mutant KRAS protein-ligand complexes, showing binding scores, key residues and interaction distances (Å).**
(DOCX)

**S3 Fig. (2D) schematic representation illustrating critical interactions between the protein-ligand, highlighting biding site residues and interaction types.**
(DOCX)

## Author contributions

**Conceptualization:** Saqib Ishaq.

**Data curation:** Saqib Ishaq.

**Methodology:** Saqib Ishaq.

**Supervision:** Saqib Ishaq, Irshad Ur Rehman, Abdul Aziz, Yasir Ali, Ajaz Ahmad, Amin Ullah.

**Validation:** Saqib Ishaq, Yasir Ali.

**Visualization:** Saqib Ishaq, Aizaz Ali, Yasir Ali.

**Writing – original draft:** Saqib Ishaq, Aizaz Ali, Obaid Habib, Shabir Ahmad Usmani, Yasir Ali.

**Writing – review & editing:** Saqib Ishaq, Aizaz Ali, Obaid Habib, Shabir Ahmad Usmani, Abdul Aziz, Yasir Ali, Amin Ullah.

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
