## [Decision Letter · Decision Letter 0]

29 Sep 2025

Dear Dr. Ishaq,

Thank you for submitting your manuscript to PLOS ONE. After careful consideration, we feel that it has merit but does not fully meet PLOS ONE’s publication criteria as it currently stands. Therefore, we invite you to submit a revised version of the manuscript that addresses the points raised during the review process.

We look forward to receiving your revised manuscript.

Kind regards,

Rituraj Purohit, Ph.D.

Academic Editor

PLOS ONE

Journal Requirements:

“yes”

“Researchers Supporting Project Number (RSP2025R350), King Saud University, Riyadh, Saudi Arabia”

Reviewers' comments:

Reviewer's Responses to Questions

**Comments to the Author**

1. Is the manuscript technically sound, and do the data support the conclusions?

Reviewer #1: Yes

Reviewer #2: Yes

Reviewer #3: Yes

2. Has the statistical analysis been performed appropriately and rigorously?

Reviewer #1: No

Reviewer #2: Yes

Reviewer #3: No

3. Have the authors made all data underlying the findings in their manuscript fully available?

Reviewer #1: No

Reviewer #2: No

Reviewer #3: Yes

4. Is the manuscript presented in an intelligible fashion and written in standard English?

Reviewer #1: No

Reviewer #2: Yes

Reviewer #3: Yes

Reviewer #1: 1. The background should better highlight the clinical significance of KRAS mutations and why nsSNP analysis is crucial compared to other oncogenic drivers.

2. Clarify the novelty of this study by distinguishing it from previous KRAS in-silico mutation studies.

3. The abstract should explicitly mention the cancer types most linked to the identified variants for translational relevance.

4. The methods section requires more details on databases and prediction tools used, including their versions and threshold criteria.

5. Justify the selection of 173 nsSNPs, were they all available in dbSNP or filtered by frequency/pathogenicity?

6. Explain why only four deleterious variants were highlighted, and whether others were borderline pathogenic.

7. Provide structural rationale (location in functional domains, switch regions, GTP-binding sites) for the identified variants.

8. Include quantitative details from molecular dynamics (RMSD, RMSF, hydrogen bonds, free energy) in the abstract or results for stronger impact. Here, I am suggesting some articles to see how to incorporate and validate the computational data: PMID: 27699663; PMID: 22974711; PMID: 24146998

9. Docking should specify the ligands used (GTP, GDP, or effector proteins) and how scoring was validated.

10. Discuss potential effects of mutations on post-translational modifications with mechanistic examples rather than listing.

11. The protein–protein interaction analysis should identify specific partners affected (RAF, PI3K, RALGDS).

12. Add a comparison with known clinically reported KRAS mutations (G12D, G12V, Q61H) to place results in context.

13. Figures should include structural visualization of mutant vs. wild type conformations for clarity.

14. Language polishing is needed (“revelled” to “revealed,” “conceivable impairment” to “potential impairment”).

15. The conclusion should stress translational applications, such as guiding drug design or prioritizing variants for functional assays.

Reviewer #2: The study used a comprehensive computational workflow to screen 173 KRAS nsSNPs and pinpoint four variants predicted to be most damaging. Detailed structural modeling and molecular dynamics revealed that these mutations destabilize KRAS and may drive oncogenic activation. Authors need to address follwing points to improve the overall study:

1. In Figure 2 it seems like nSNP has zero number. Authors need to find beter way for the representation.

2. Reporting an RMSD of exactly 0.000 Å is not realistic; clarify what this value represents or correct if it is an artifact of the software.

3. The list of interacting residues is useful, but it would be more convincing to include a concise table summarizing hydrogen-bond distances and key hydrophobic contacts for each ligand.

4. Authors have mention that the each compound has “one violation of Lipinski’s rule,” but do not state which rule is violated.

5. It would strengthen the paper to discuss how the predicted phosphorylation or ubiquitylation sites might influence KRAS activity or drug binding.

6. Relate your computational findings to existing experimental data on salirasib or similar inhibitors to give the work broader relevance.

7. In 3.1 you state that 173 nsSNPs were identified, but in 3.2 you mention “All 127 nsSNPs recruited from dbSNP.” Please clarify whether 173 or 127 were analyzed.

8. Figure 2 is described as “Percentage of all SNPs,” but the y-axis numbers (12000, 10000, etc.) suggest absolute counts. Either relabel the axis or revise the legend.

9. In the conservation analysis, discuss why G138E being “exposed and functionally significant” is important for KRAS activity.

10. Authors can citre recent literature to improve the discussion of MS. Here are few suggetsed articles:PMID: 34365920, PMID: 33100208, PMID: 37116071.

11. Clarify whether the TM-scores (>0.94) imply that all mutants retain the overall KRAS fold, and comment on the biological significance of RMSD values around 1–1.7 Å.

12. “carried out in sveral steps, for” Typos error in the statement.

Reviewer #3: 1. The manuscript refers to several different simulation times (e.g., 2 µs in some sections and 100 ns in others). State clearly which systems were simulated for which durations, whether different runs are independent replicates or different systems.

2. Docking score differences of ~0.1–0.3 kcal·mol⁻¹ are within scoring noise. Include redocking of any available co-crystallized ligands, an ensemble of receptor conformations, and/or consensus rescoring or decoy sets to strengthen claims that salirasib is superior.

3. For KRAS, explicitly state whether GDP/GTP and Mg²⁺ were present and how their parameters were treated, these strongly affect KRAS conformation and ligand binding.

4. The author should explain how ligand charges/parameters were generated. Provide validation (geometry/charge). Poor parameterization can bias both docking and MD results.

5. Single MD trajectories are insufficient to claim robust differences. Run at least 3 independent replicates (different initial velocities) per system, report averages SD for RMSD, and show that observed differences are reproducible.

**Do you want your identity to be public for this peer review?** For information about this choice, including consent withdrawal, please see our Privacy Policy

Reviewer #1: No

Reviewer #2: No

Reviewer #3: No

---

## [Author Response · Author response to Decision Letter 1]

10 Dec 2025

Dear Editor,

Thank you very much for the reviews of this manuscript. We have described below our responses to the comments from reviewer 1, 2 and 3, followed the suggestions of reviewers and editor. The changes made in the revised manuscript are highlighted yellow.

---

## [Decision Letter · Decision Letter 1]

5 Jan 2026

Pathogenic KRAS Variants Disrupt Structure and Dynamics: Insights from Integrated Computational Analyses

PONE-D-25-48831R1

Dear Dr. Ishaq,

We’re pleased to inform you that your manuscript has been judged scientifically suitable for publication and will be formally accepted for publication once it meets all outstanding technical requirements.

Kind regards,

Rituraj Purohit, Ph.D.

Academic Editor

PLOS One

Additional Editor Comments (optional):

Reviewers' comments:

Reviewer's Responses to Questions

**Comments to the Author**

Reviewer #1: All comments have been addressed

Reviewer #2: All comments have been addressed

2. Is the manuscript technically sound, and do the data support the conclusions?

Reviewer #1: Yes

Reviewer #2: Yes

3. Has the statistical analysis been performed appropriately and rigorously?

Reviewer #1: Yes

Reviewer #2: Yes

4. Have the authors made all data underlying the findings in their manuscript fully available?

Reviewer #1: Yes

Reviewer #2: Yes

5. Is the manuscript presented in an intelligible fashion and written in standard English?

Reviewer #1: Yes

Reviewer #2: Yes

Reviewer #1: The authors have addressed all the queries raised by the reviewers and the manuscript is now acceptable.

Reviewer #2: The authors have thoroughly addressed all the comments, incorporating the suggested revisions and improving the overall clarity and quality of the manuscript. It is now ready to be published in current form.

**Do you want your identity to be public for this peer review?** For information about this choice, including consent withdrawal, please see our Privacy Policy

Reviewer #1: No

Reviewer #2: No

---

## [Editor Report · Acceptance letter]

PONE-D-25-48831R1

PLOS One

Dear Dr. Ishaq,

I'm pleased to inform you that your manuscript has been deemed suitable for publication in PLOS One. Congratulations! Your manuscript is now being handed over to our production team.

Kind regards,

on behalf of

Dr. Rituraj Purohit

Academic Editor

PLOS One